# Association of Insomnia with 30-Day Postpartum Readmission: A Retrospective Analysis

**DOI:** 10.3390/ijerph20115955

**Published:** 2023-05-25

**Authors:** Anthony M. Kendle, Justin Swanson, Jason L. Salemi, Judette M. Louis

**Affiliations:** 1Department of Obstetrics and Gynecology, Morsani College of Medicine, University of South Florida, Tampa, FL 33602, USA; jsalemi@usf.edu (J.L.S.); jlouis1@usf.edu (J.M.L.); 2College of Public Health, University of South Florida, Tampa, FL 33612, USA; jswanson1@usf.edu

**Keywords:** readmission, postpartum, severe maternal mortality, obstetric comorbidity index, insomnia, sleep, National Readmission Database (NRD), Healthcare Cost and Utilization Project (HCUP)

## Abstract

Insomnia is prevalent in pregnancy and is associated with increased use of health services. We aimed to evaluate the association between insomnia diagnosed at the delivery hospitalization and risk of 30-day postpartum readmission. We conducted a retrospective analysis of inpatient hospitalizations from the 2010–2019 Nationwide Readmissions Database. The primary exposure was a coded diagnosis of insomnia at delivery as determined by ICD-9-CM and ICD-10-CM codes. Obstetric comorbidities and indicators of severe maternal morbidity were also determined through coding. The primary outcome was all-cause 30-day postpartum readmission. Survey-weighted logistic regression was used to generate crude and adjusted odds ratios representing the association between maternal insomnia and postpartum readmission. Of over 34 million delivery hospitalizations, 26,099 (7.6 cases per 10,000) had a coded diagnosis of insomnia. People with insomnia experienced a 3.0% all-cause 30-day postpartum readmission rate, compared to 1.4% among those without insomnia. After controlling for sociodemographic, clinical, and hospital-level factors, insomnia was associated with 1.64 times higher odds of readmission (95% CI 1.47–1.83). After adjustment for obstetric comorbidity burden and severe maternal morbidity, insomnia was independently associated with 1.33 times higher odds of readmission (95% CI 1.18–1.48). Pregnant patients with insomnia have higher rates of postpartum readmission, and diagnosis of insomnia is independently associated with increased odds of readmission. Additional postpartum support may be warranted for pregnancies affected by insomnia.

## 1. Introduction

Readmission after obstetric hospitalization and how it relates to quality of maternity care in the United States is a topic of growing interest [1]. Large, multi-state cohorts have demonstrated that postpartum readmission rates are increasing: one study of nearly 6 million deliveries demonstrated a 44% increase in postpartum readmission from 2004 to 2010 [2]. Additionally, data from the United States’ Nationwide Readmissions Database reported 1.06% of all deliveries required readmission in 2013 [3]. High rates of readmission come at a cost: In 2018, the average cost of 30-day hospital readmission in the non-obstetric population was estimated at USD 15,200 [4]. While the economic burden of obstetric readmission is less studied, large, administrative data sets capturing 30-day postpartum readmission complicated by severe maternal morbidity (SMM) estimated a mean cost of USD 47,480 per patient [5]. One must also consider the potential adverse effect of readmission on maternal–infant bonding.

As such, there is a body of literature assessing how various diagnoses during pregnancy are associated with postpartum readmission. Insomnia is the most common sleep disorder affecting pregnant and non-pregnant adults [6,7,8,9]. While insomnia has been associated with adverse maternal and neonatal outcomes in pregnancy [10,11,12], studies suggest the time of greatest sleep disturbance occurs in the postpartum period [13,14]. Moreover, there is an established relationship with poor sleep and subsequent development of postpartum depression [14,15]. Together, these features of insomnia are well poised to influence maternal health after delivery, and there are no existing data examining the effect of sleep disturbance and need for rehospitalization.

Using a large, nationally representative database, we aimed to estimate the association between a coded diagnosis of insomnia during pregnancy and the risk of 30-day postpartum hospital readmission over a ten-year period from 2010 to 2019. Additionally, we aimed to assess the interaction among insomnia, maternal comorbid conditions, and SMM in their contributions to postpartum readmission and the principal diagnoses driving readmission. We hypothesized that patients that receive a coded diagnosis of insomnia will experience higher rates of readmission.

## 2. Materials and Methods

### 2.1. Data Source

This study is retrospective analysis of inpatient hospitalizations from 2010 to 2019 using data from the Nationwide Readmissions Database (NRD) [16], developed for the Healthcare Cost and Utilization Project (HCUP). The NRD is the largest nationally representative, publicly available, all-payer readmissions database in the United States. In 2019, the NRD drew from 30 geographically dispersed states encompassing 61.8% of the U.S. population and 60.4% of all inpatient hospitalizations; this covers 18 million unweighted (35 million weighted) annual discharges [16]. The sampling frame for the NRD includes discharges from community hospitals (excluding rehabilitation or long-term acute care facilities), and post-stratification for weighting occurs by several hospital and patient characteristics. Because the NRD contains completely de-identified patient data, this study was deemed exempt by the University of South Florida Institutional Review Board.

### 2.2. Study Sample and Measures

The study sample was generated by first identifying index events, which were defined as a hospitalization to a female patient aged 15–49 years during which a delivery occurred. The NRD captures up to 40 diagnostic and 25 procedural codes associated with each hospitalization. Before 1 October 2015, International Classification of Diseases, Ninth Revision, Clinical Modification (ICD-9-CM) codes were used; after this date, the Tenth Revision (ICD-10-CM) was used. Delivery encounters were identified using a combination of diagnosis-related group (DRG) codes and ICD-9CM [17] or ICD-10-CM [18] codes. Since the NRD does not link patients across calendar years, index events excluded hospitalizations during the month of December to allow for 30 days of follow-up time. When maternal death occurred prior to discharge from the delivery hospitalization, we excluded these records from the study since the outcome (postpartum readmission) cannot be observed.

The primary outcome examined in this study was a binary indicator of all-cause readmission to any facility within 30 days of discharge from the delivery hospitalization. Although hospitalizations for the same individual are linked across different facilities, the NRD would not pick up hospitalizations for the same person that occur in facilities located in different states. Overall and for various population subgroups, the 30-day postpartum readmission rate was calculated as the number of deliveries for which at least one 30-day all-cause readmission occurred divided by the total number of index (delivery) hospitalizations.

The primary exposure of interest was insomnia, which was captured as a binary indicator of insomnia-related diagnostic codes (ICD-9: 307.40–307.42, 327.00–327.09, 780.51–780.52, V69.4–V69.5; ICD-10: F51.01–F51.09, F51.12, F51.9, G47.00–G47.09, Z72.82, Z73.81) as documented during the delivery hospitalization. The selection of insomnia-related diagnosis codes was based on the International Classification of Sleep Disorders [19] and the Handbook of Sleep Medicine [20]. SMM was coded as a single binary indicator of any of the 21 conditions that are part of the definition proposed by the Centers for Disease Control and Prevention [21]. Additionally, obstetric comorbidity scores were calculated for index events using the scoring system developed by Leonard et al. to predict SMM from hospital discharge data [22]. Age was categorized in years as 15–19, 20–24, 25–29, 30–34, 35–39, and 40–49. The primary payer of the hospitalization (i.e., patient’s source of insurance) was categorized as either government (Medicare/Medicaid), private, or other (self-pay, no charge, or other methods of payment). Although the NRD does not contain patient-level income, it does provide the household income quartile of the patient’s residential ZIP code. Severity of illness during the index event was determined using a proprietary algorithm incorporating patient clinical and demographic characteristics [23]. Hospital-level factors include hospital type (urban teaching, urban non-teaching, or rural) and bed size (small, medium, or large).

### 2.3. Statistical Analysis

Descriptive statistics including frequencies and percentages were used to characterize the distribution of sociodemographic and clinical characteristics among delivery hospitalizations in the US. Survey-weighted logistic regression was used to calculate odds ratios (OR) and 95% confidence intervals (CI) estimating the association between insomnia and 30-day all-cause postpartum readmission. In addition to unadjusted models, two multivariable models were created. The first multivariable model included all sociodemographic and clinical covariates except for the indication of SMM at delivery and obstetric comorbidity index score. All study variables were included in the second multivariable model. Because blood transfusions comprise a disproportionately high proportion of SMM events, sensitivity analyses were performed using a definition of SMM that required documentation of an SMM-related condition other than blood transfusion.

Primary reasons for postpartum readmission were explored using the principal diagnosis of the first readmission event occurring within 30 days of discharge from the delivery hospitalization. Individual ICD-9-CM/ICD-10-CM principal diagnosis codes were classified into clinically meaningful categories with the Clinical Classifications Software (CCS) tool developed by HCUP [24,25]. For CCS categories in which the reason for readmission was non-specific (e.g., other complications of birth, puerperium affecting management of mother, other complications of pregnancy, or other), we cross-referenced individual ICD-9-CM/ICD-10-CM codes and reassigned indication for readmission into the following categories: hypertensive disorders, infectious complications, mental health disorders, hematologic disorders, surgical complications, cardiovascular disorders, pulmonary disorders, venous thromboembolism, neurologic disorders, endocrine disorders, hepatobiliary disorders, and other/not specified. The frequency and prevalence of insomnia were calculated for each major CCS category. Missing values for variables were not included in counts and calculations; thus, some subgroups may not total to the total study population. Counts based on ten or fewer events were suppressed in accordance with the HCUP data suppression rules. All statistical analyses were performed using R (version 4.1.2) (R Core Team: Vienna, Austria) [26].

## 3. Results

During the study period, there were more than 34 million delivery hospitalizations that qualified as index events (Table 1). Of these, 26,099 had an associated diagnosis of insomnia, a prevalence of 7.6 cases of insomnia per 10,000 delivery hospitalizations. Patients with insomnia were more likely to have SMM (4.4% vs. 1.7%) and had higher obstetric comorbidity scores (27.4 ± 24.6 vs. 9.6 ± 15.0).

People whose delivery hospitalizations had an insomnia diagnosis experienced a 30-day all-cause readmission rate of 3.0% (95% CI 2.7–3.4%), which was a 2.26 (95% CI 2.03–2.53) times higher odds of readmission when compared to deliveries without a diagnosis of insomnia (Table 2). After controlling for sociodemographic, clinical, and hospital factors, people with insomnia experienced 1.64 (95% CI 1.47–1.83) times higher odds of postpartum readmission. Even after further adjustment for the obstetric comorbidity burden and SMM at delivery, insomnia was independently associated with 1.33 (95% CI 1.18–1.48) times higher odds of readmission. Substituting indicators of SMM with transfusion-exclusionary counterparts reduced the frequency of SMM and mean obstetric comorbidity scores nearly by half (Appendix A) but did not substantially change the association between insomnia and 30-day readmission (Appendix A).

Reasons for readmission after a diagnosis with insomnia are presented in Figure 1. Of the 791 readmissions captured among patients with insomnia, the most common principal diagnoses for readmission were hypertensive disorders (*n* = 206, 26.0%), infectious complications (*n* = 181, 22.9%), mental health disorders (*n* = 97, 12.3%), and hematologic disorders (*n* = 50, 6.3%). Among all 30-day postpartum readmissions, insomnia was highly prevalent among those readmitted for mental health disorders (117.2 cases per 10,000 readmissions), pulmonary disorders (66.5 cases per 10,000), and neurologic disorders (49.3 cases per 10,000). Table 3 shows the odds of readmission for index hospitalizations affected by insomnia based on individual readmissions classifications.

Top principal Clinical Classification Software diagnosis categories for readmissions occurring after index hospitalizations with a coded diagnosis of insomnia. Light gray bars indicate frequency of each diagnosis category among all readmissions with a coded diagnosis of insomnia. Dark gray bars indicate the prevalence of insomnia (per 10,000) within each readmission category among all readmissions. Index hospitalizations include delivery hospitalizations among patients aged 15–49 years who were at risk for 30-day readmission. In accordance with data suppression rules, counts based on ten or fewer events were suppressed and recategorized under “Other/Not specified”.

## 4. Discussion

The results of our study explore the relationship between prenatally diagnosed insomnia and 30-day postpartum readmission in the United States from 2010 to 2019. We found that a coded diagnosis of insomnia was associated with a greater than two-fold increased odds of readmission compared to patients without this diagnosis. Patients with insomnia were more likely to experience severe illness and have comorbidities related to severe maternal outcomes. Even after controlling for these factors, a diagnosis of insomnia was independently associated with increased odds of readmission following delivery hospitalization.

In a separate analysis of the NRD, Clapp et al. defined primary indications for postpartum readmission during the year 2013 [2]. Among the most common indications were hypertension (21.6%), wound infection (13.0%), and hemorrhage (5.8%). Our cohort with insomnia encountered many of these same common indications for readmission with one key difference: mental health disorders were more commonly implicated as reason for readmission at a rate of 12.3%, compared to 2.5% in other large cohorts [3]. There is a strong association of insomnia with mental health disease. Meta-analysis data of adult patients with insomnia has shown a 2.10 (95% CI 1.86–2.38) increased odds of comorbid depression [27]. A population-based study of pregnant patients identified a 61.9% and 14.6% prevalence of insomnia and depressive symptoms, respectively, among patients in the third trimester with strong correlation between the two diagnoses [28]. Furthermore, in postpartum patients with depression, poor sleep increases frequency of thoughts of self-harm (OR 1.15, 95% CI 1.02–1.29) [29]. In our study, the highest prevalence of insomnia was noted in readmissions for a mental health indication compared to any other readmission category.

Our findings offer insight into a theoretical framework by which a prenatal diagnosis of insomnia drives postpartum readmission. Existing insomnia is expected to worsen in the postpartum period with the additional sleep disturbances related to caring for a newborn. While postpartum worsening of mental health conditions has been clearly established, sleep disturbance and short sleep duration have also been associated with increased levels of inflammatory biomarkers such as C-reactive protein and interleukin-6 [30]. By this mechanism, increased inflammation and immunologic modulation may drive readmissions for hypertension, infection, and thromboembolic disorders. Given the biologic basis for insomnia to negatively affect maternal health, additional clinical attention to postpartum sleep quality in addition to routine screening for postpartum depression may provide opportunity for intervention and ultimately reduce healthcare utilization.

As this study relies on the use of a large, administrative database, there are several important limitations that must be considered. Identification of the primary exposure in this study was reliant on ICD-9 and ICD-10 diagnostic codes and thus is subject to coding error and diagnostic bias. The American Academy of Sleep Medicine has defined specific diagnostic criteria when evaluating a patient for insomnia [31]. In general, sleep disorders tend to be under coded in pregnancy, and we recognized that the primary exposure in this study is underrepresented. As such, those that received a coded diagnosis of insomnia in this study were likely among those who had an existing or chronic diagnosis, presented with severe or refractory symptomatology, or required behavioral therapy or pharmacotherapy during pregnancy. The results of this study may better reflect the effect of severe insomnia on postpartum readmission. Capture of secondary exposures and comorbidities may have been similarly limited. While regression models in this study were robust in our ability to control for SMM and obstetric comorbidities, effect modification of other comorbid conditions—particularly specific mental health diagnoses—is not fully explored.

Because the NRD does not report race and ethnicity, we are unable to assess how these populations are differentially affected by insomnia or the rate of readmission. Racial discrimination is a strong mediator of insomnia in non-pregnant populations and contributes to racial sleep inequity [32]. Furthermore, non-Hispanic black people have a 20% increased risk of encountering non-blood transfusion SMM at time of delivery compared to non-Hispanic white people [33]. Given the existing literature, we expect similar racial disparities in sleep health as it relates to postpartum readmission, and this topic deserves investigation in future studies.

In addition to study exposures, identification of readmission diagnosis is limited by the study design. For each 30-day readmission, only the principal diagnosis for readmission was captured and analyzed. As such, secondary diagnoses of patients readmitted with multiple acute complications are not fully represented. Thus, the true distribution of readmission diagnoses may be misrepresented due to non-specific coding. Lastly, the nature of hospital linkage does not capture complications resulting in maternal death that occurred after delivery hospitalization.

Despite these limitations, our study has several notable strengths. With ten years of data, we leveraged the large sample size and statistical power of the NRD to examine rare exposures such as insomnia during pregnancy. To our knowledge, this is the first and largest study to explore the relationship between insomnia and odds of readmission after delivery.

## 5. Conclusions

The independent association between our primary exposure and outcome draws attention to the importance of screening pregnant and postpartum patients for poor sleep. For patients with severe insomnia or those receiving treatment, our findings also support a role for closer postpartum interval follow up with special attention to mental health and sleep health services. Future prospective studies are needed to better quantify the maternal risk of insomnia in the postpartum period and to assess the degree to which treatment of insomnia can reduce postpartum healthcare utilization.

## Figures and Tables

**Figure 1 ijerph-20-05955-f001:**
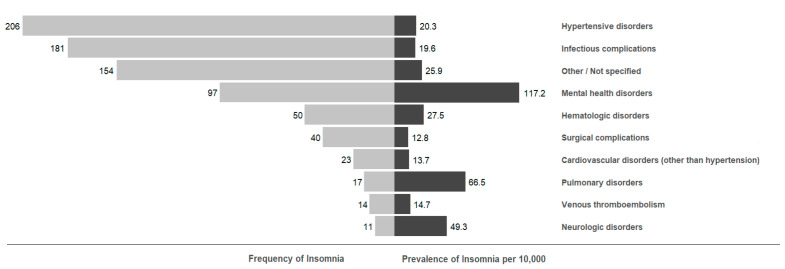
Primary Reasons for Readmission after Index Hospitalizations with a Coded Diagnosis of Insomnia, Nationwide Readmissions Database, 2010–2019.

**Table 1 ijerph-20-05955-t001:** Distribution of Sociodemographic, Clinical, and Hospital Characteristics Among Index ^a^ Hospitalizations with and without a Coded Diagnosis of Insomnia, Nationwide Readmissions Database, 2010–2019.

Characteristic	No Insomnia		Insomnia	
	n ^b^	% ^c^	n ^b^	% ^c^
**All Discharges**	34,283,319	100.0	26,099	100.0
**Age (years)**				
15–19	2,332,004	7.0	979	4.0
20–24	7,361,734	22.1	3901	15.8
25–29	9,851,165	29.6	6864	27.8
30–34	9,215,203	27.7	8118	32.8
35–39	4,488,095	13.5	4859	19.7
40–49	1,035,117	3.0	1377	5.3
**Household income ^d^**				
Highest quartile	7,334,019	21.6	6160	23.8
Third quartile	8,529,700	25.1	6876	26.6
Second quartile	8,602,286	25.3	6362	24.6
Lowest quartile	9,511,438	28.0	6454	25.0
**Primary payer ^e^**				
Private	17,892,600	52.3	13,911	53.4
Government	14,720,829	43.0	10,977	42.1
Other	1,601,520	4.7	1181	4.5
**Elective admission**				
Non-elective	17,229,921	50.3	13,610	52.2
Elective	17,010,816	49.7	12,458	47.8
**Timing of admission**				
Weekday	27,644,201	80.6	21,182	81.2
Weekend	6,639,118	19.4	4917	18.8
**Severity of illness ^f^**				
Minor loss of function	21,049,316	61.4	7257	27.8
Moderate loss of function	10,981,063	32.0	13,272	50.9
Major loss of function	2,187,426	6.4	5286	20.3
Extreme loss of function	62,018	0.2	279	1.1
**Hospital location/type**				
Urban teaching	20,747,627	60.5	18,293	70.1
Urban non-teaching	10,050,613	29.3	5612	21.5
Rural	3,485,078	10.2	2194	8.4
**Hospital bed size**				
Small	4,751,918	13.9	4238	16.2
Medium	9,409,904	27.4	6404	24.5
Large	20,121,497	58.7	15,457	59.2
**Severe maternal morbidity ^g^**				
No	33,693,758	98.3	24,946	95.6
Yes	589,561	1.7	1152	4.4
**Obstetric comorbidity score ^h^, Mean (SD)**	9.6 (15.0)		27.4 (24.6)	

Abbreviation: SD, Standard Deviation: ^a^ Index events include delivery hospitalizations among patients aged 15–49 years who were at risk for 30-day readmission. ^b^ Weighted to estimate national frequencies. The sum of subgroups may not add up to the total due to missing data. ^c^ Column percentages may be used to compare the distribution of study characteristics in patients with and without insomnia. ^d^ Determined by median household income for patient’s zip code. ^e^ Government includes Medicare and Medicaid. Other includes self-pay, no charge, and other payers. ^f^ All patient refined diagnosis related group illness severity is estimated using a proprietary algorithm incorporating patient sociodemographics, comorbidities, and services rendered [23]. ^g^ Severe maternal morbidity includes any of the 21 conditions captured by the Centers for Disease Control and Prevention [21]. ^h^ Obstetric comorbidity scores were calculated using a scoring system developed to predict severe maternal morbidity from hospital discharge data [22].

**Table 2 ijerph-20-05955-t002:** Sociodemographic, Clinical, and Hospital Characteristics Associated with 30-Day All-Cause Readmission Among Index ^a^ Hospitalizations, Nationwide Readmissions Database, 2010–2019.

Characteristic	Readmission Rate ^b^, % (95% CI)	Odds Ratio (95% CI) ^c^		
		Crude Model	Adjusted ^d^ Model 1	Adjusted ^e^ Model 2
**All Discharges**	1.36 (1.34–1.39)			
**Age (years)**				
15–19	1.44 (1.40–1.48)	1.15 (1.12–1.18)	1.00 (0.98–1.02)	1.05 (1.03–1.07)
20–24	1.32 (1.29–1.35)	1.06 (1.04–1.07)	0.99 (0.98–0.96)	1.00 (0.98–1.01)
25–29	1.25 (1.23–1.28)	Reference	Reference	Reference
30–34	1.29 (1.27–1.32)	1.03 (1.02–1.05)	1.09 (1.08–1.11)	1.07 (1.06–1.09)
35–39	1.61 (1.57–1.65)	1.29 (1.27–1.31)	1.33 (1.31–1.35)	1.24 (1.22–1.26)
40–49	2.11 (2.04–2.18)	1.70 (1.66–1.74)	1.66 (1.62–1.70)	1.50 (1.46–1.54)
**Household income ^f^**				
Highest quartile	1.15 (1.10–1.19)	Reference	Reference	Reference
Third quartile	1.29 (1.26–1.32)	1.13 (1.11–1.15)	1.12 (1.10–1.14)	1.10 (1.08–1.12)
Second quartile	1.37 (1.34–1.41)	1.20 (1.18–1.23)	1.18 (1.15–1.20)	1.15 (1.13–1.17)
Lowest quartile	1.58 (1.54–1.63)	1.39 (1.36–1.42)	1.29 (1.26–1.31)	1.24 (1.21–1.26)
**Primary payer ^g^**				
Private	1.18 (1.15–1.21)	Reference	Reference	Reference
Government	1.60 (1.57–1.63)	1.36 (1.34–1.38)	1.33 (1.31–1.35)	1.28 (1.26–1.29)
Other	1.21 (1.16–1.27)	1.03 (1.00–1.06)	1.05 (1.02–1.08)	1.04 (1.01–1.07)
**Elective admission**				
Non-elective	1.31 (1.05–1.58)	Reference	Reference	Reference
Elective	1.41 (1.37–1.45)	0.95 (0.94–0.97)	1.02 (1.01–1.04)	1.02 (1.01–1.04)
**Timing of admission**				
Weekday	1.39 (1.36–1.42)	Reference	Reference	Reference
Weekend	1.25 (1.23–1.28)	0.90 (0.89–0.91)	0.90 (0.89–0.91)	0.92 (0.91–0.93)
**Severity of illness ^h^**				
Minor loss of function	1.01 (1.00–1.03)	Reference	Reference	Reference
Moderate loss of function	1.64 (1.60–1.67)	1.62 (1.60–1.64)	1.59 (1.57–1.61)	1.36 (1.35–1.38)
Major loss of function	3.17 (3.08–3.27)	3.20 (3.15–3.25)	2.99 (2.95–3.04)	1.73 (1.69–1.76)
Extreme loss of function	7.40 (7.02–7.78)	7.80 (7.42–8.19)	7.01 (6.67–7.37)	1.97 (1.86–2.08)
**Hospital location/type**				
Urban teaching	1.46 (1.42–1.50)	Reference	Reference	Reference
Urban non-teaching	1.21 (1.18–1.24)	0.83 (0.81–0.85)	0.91 (0.89–0.93)	0.94 (0.93–0.96)
Rural	1.22 (1.18–1.25)	0.83 (0.81–0.85)	0.86 (0.84–0.88)	0.89 (0.86–0.91)
**Hospital bed size**				
Small	1.23 (1.17–1.30)	Reference	Reference	Reference
Medium	1.31 (1.27–1.35)	1.07 (1.03–1.10)	1.05 (1.02–1.08)	1.05 (1.02–1.09)
Large	1.42 (1.38–1.46)	1.15 (1.11–1.19)	1.11 (1.07–1.14)	1.10 (1.07–1.14)
**Insomnia**				
No	1.36 (1.34–1.39)	Reference	Reference	Reference
Yes	3.03 (2.69–3.38)	2.26 (2.03–2.53)	1.64 (1.47–1.83)	1.33 (1.18–1.48)
**Severe maternal morbidity ^i^**				
No	1.31 (1.28–1.33)	Reference	N/A	Reference
Yes	4.56 (4.43–4.70)	3.61 (3.54–3.69)	N/A	1.76 (1.72–1.81)
**Obstetric comorbidity score ^j^ (10 points)**		1.252 (1.250–1.256)	N/A	1.162 (1.158–1.165)

Abbreviation: CI, Confidence Interval; N/A, Not Applicable: ^a^ Index events include delivery hospitalizations among patients aged 15–49 years who were at risk for 30-day readmission. ^b^ Readmission rates calculated as the percentage of index hospitalizations readmitted within 30 days. Frequencies were weighted to estimate national readmission rates. ^c^ Odds ratios and 95% confidence intervals were generated using survey-weighted logistic regression. ^d^ Adjusted model 1 adjusts for all covariates included in the table except for severe maternal morbidity and obstetric comorbidity score. ^e^ Adjusted model 2 adjusts for all covariates included in the table. ^f^ Determined by median household income for patient’s zip code. ^g^ Government includes Medicare and Medicaid. Other includes self-pay, no charge, and other payers. ^h^ All patient refined diagnosis related group illness severity is estimated using a proprietary algorithm incorporating patient sociodemographics, comorbidities, and services rendered [23]. ^i^ Severe maternal morbidity includes any of the 21 conditions captured by the Centers for Disease Control and Prevention [21]. ^j^ Obstetric comorbidity scores were calculated using a scoring system developed to predict severe maternal morbidity from hospital discharge data [22]. Odds ratios reflect a 10-point change in score.

**Table 3 ijerph-20-05955-t003:** Odds of Readmission after Index Hospitalizations with a Coded Diagnosis of Insomnia, Nationwide Readmissions Database, 2010–2019.

Clinical Class Software Category *	Odds Readmission (95% Confidence Interval)
Hypertensive disorders	1.35 (1.09–1.66)
Infectious complications	1.75 (1.40–2.20)
Other/Not specified	1.94 (1.52–2.45)
Mental health disorders	8.43 (5.73–12.40)
Hematologic disorders	2.15 (1.38–3.35)
Surgical complications	1.31 (0.75–2.28)
Cardiovascular disorders	0.93 (0.50–1.74)
Pulmonary disorders	3.69 (1.54–8.83)
Venous thromboembolism	0.83 (0.32–2.13)
Neurologic disorders	2.50 (0.86–7.27)

* Version: CCSR v2023.1.

## Data Availability

Data used for this study is publicly available as part of the Healthcare Cost and Utilization Project available at: https://hcup-us.ahrq.gov (accessed on 22 May 2023).

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
