# Peer review of "Association of Insomnia with 30-Day Postpartum Readmission: A Retrospective Analysis"

_ijerph, 2023, doi:10.3390/ijerph20115955_

Round 1

Reviewer 1 Report

This is a very interesting article, addressing a topic that has an increasing impact on public health. The scientific quality is very good and there are only a few minor details that can help to further increase its accuracy:

- lines 43 to 45, it may be interesting to briefly state the prevalence of the other causes of postpartum readmission

- line 56, the acronym (SMM) has already been explained in line 40

- line 211, the reference number (Clapp et al) is missing

There being nothing further to point out, I recommend acceptance after these minor revisions

Author Response

- lines 43 to 45, it may be interesting to briefly state the prevalence of the other causes of postpartum readmission
The prevalence of other common reasons for postpartum admission are discussed in the second paragraph of the discussion. However, we have added percentages so that the reader can understand our results in the context of other indications for readmission (lines 214-215)

- line 56, the acronym (SMM) has already been explained in line 40
Thank you. This has been corrected.

- line 211, the reference number (Clapp et al) is missing
We apologize for this oversight. This references has been added to the text.

Reviewer 2 Report

This study used a large sample size and ten years of population data to determine the relationship between insomnia and the 30-day postpartum readmission rate. The association between insomnia and a higher rate of postpartum readmission is established independently of obstetric comorbidity and severe maternal morbidity. I recommend accepting the paper with some minor suggestions.

1.    In the statistical analysis, the p value can be added.

2.    The top 3 bars have the same heights, but values are missing and different.

3.    Among the primary reasons for readmission of patients with insomnia, it will also be interesting to show the detailed association between these factors and readmission to reveal the direct exposure of interest further.

Author Response

  1. In the statistical analysis, the p value can be added.

The goal of our “Table 1” is to simply describe the distribution of various characteristics across delivery hospitalizations with and without a coded diagnosis of insomnia. The sample size of the NRD is so large that all differences end up approaching statistical significance. Moreover, since our goal in including these characteristics in our analysis is ultimately to remove their contribution to the bias that confounds the association between insomnia and readmission. As such, we prefer not to include p-values in an effort to avoid overinterpreting these crude associations. If the reviewer or editor still see value in including p-values, we are happy to include.

  1. The top 3 bars have the same heights, but values are missing and different.

It appears the figure was incorrectly cropped on the original submission. It has been replaced. The values for the top three bars are 205, 181, and 154 from top to bottom.

  1. Among the primary reasons for readmission of patients with insomnia, it will also be interesting to show the detailed association between these factors and readmission to reveal the direct exposure of interest further.

Table 3 has been added to the manuscript. This outlines the odds of readmission after index hospitalizations with a coded diagnosis of insomnia for each readmission category.

Reviewer 3 Report

Dear authors

I would like to thank you for giving me the opportunity to review the manuscript entitled “Association of Insomnia with 30-Day Postpartum Readmission”. This study as a retrospective analysis used a large amount of data that provide valuable information regarding postpartum. The study is well-written and well-designed. I have only two minor comments:  

Title

- I think it is better to add “A retrospective analysis” to the title

Abstract

- You should provide that your study what messages have for clinical practice

Introduction

- The more and comprehensive literature review is needed

Methods

- What about ethical considerations?

- Explain how missing data were addressed

Discussion

- I would like to see more implication for practice according to the results

Author Response

Title

- I think it is better to add “A retrospective analysis” to the title

The title has been revised.

Abstract

- You should provide that your study what messages have for clinical practice 

The following has been added to the abstract: “Additional postpartum support may be warranted for pregnancies affected by insomnia.” (Lines 25-27)

Introduction

- The more and comprehensive literature review is needed 

We conducted an extensive literature review for this study as we have with other studies conducted by team members that relate to insomnia and pregnancy. There is a limited body of literature relating to insomnia during pregnancy, and even less so as it relates to the postpartum period. We have not identified additional references that would enhance this manuscript. However, we are happy to consider if there are additional and specific gaps identified by the reviewer.

Methods

- What about ethical considerations?

We appreciate the ethical conduction of research, especially as it relates to patient privacy. HCUP does not contain unique patient identifiers (instead the unit of measurement is the hospitalization) and is publicly available. Whereas ethical concerns may arise from other commercially-available databases with regard to patient privacy, we do not feel that these considerations are applicable in the methods used for this study.

- Explain how missing data were addressed

Missing values for variables were not included in counts and calculations, thus some subgroups may not total to the total study population. (Lines 139-141)

Discussion

- I would like to see more implication for practice according to the results

We have augmented the discussion to include some additional implications for clinical practice: “Given the biologic basis for insomnia to negatively affect maternal health, additional clinical attention to postpartum sleep quality in addition to routine screening for postpartum depression may provide opportunity for intervention and ultimately reduce healthcare utilization” (Lines 239-44). However, because of this study relies on administrative data, our findings should be regarded as hypothesis generating and are not provide specific information to inform specific clinical practice. 

Reviewer 4 Report

This is a very interesting study investigating the association of Insomnia with 30-Day Postpartum Readmission The paper was well-written and easy to follow.  However, there are several aspects that the authors need to look into    1. Line 187- It was stated that insomnia was highly prevalent among those readmitted for mental health disorders. At least 4% of the studied population were teenagers aged between 15-19 s. I wonder if the mental health disorders were higher among this age group.    2. I think the BMI, of the studied population may be important. (See Reference) 
Kalmbach, D. A., Cheng, P., Sangha, R., O'Brien, L. M., Swanson, L. M., Palagini, L., Bazan, L. F., Roth, T., & Drake, C. L. (2019). Insomnia, Short Sleep, And Snoring In Mid-To-Late Pregnancy: Disparities Related To Poverty, Race, And Obesity. Nature and science of sleep, 11, 301–315. Insomnia, Short Sleep, And Snoring In Mid-To-Late Pregnancy: Dispariti | NSS   Otherwise, it is a good paper. 

Author Response

  1. Line 187- It was stated that insomnia was highly prevalent among those readmitted for mental health disorders. At least 4% of the studied population were teenagers aged between 15-19 s. I wonder if the mental health disorders were higher among this age group.   

When considering the general population of index visits, a very large proportion (26.7%) of teenage readmissions were due to mental health reasons. However, when considering those index visits with insomnia, a much smaller proportion (8.2%) of teenage readmissions were due to mental health reasons. This 8.2% is within the range (4.9%-10.8%) of the other age groups, so it does not appear that the <20yo age group is largely influencing the relationship between insomnia and mental health readmissions in this analysis. Please note that due to the low proportions of insomnia, readmission, and some age groups, it is difficult to fully characterize those relationships to that level of granularity. However, we can say that overall, those with an insomnia diagnosis code upon index were several orders of magnitude more likely than the general population to have a mental health disorder as the reason for readmission.

  1. I think the BMI, of the studied population may be important. (See Reference) 
    Kalmbach, D. A., Cheng, P., Sangha, R., O'Brien, L. M., Swanson, L. M., Palagini, L., Bazan, L. F., Roth, T., & Drake, C. L. (2019). Insomnia, Short Sleep, And Snoring In Mid-To-Late Pregnancy: Disparities Related To Poverty, Race, And Obesity. Nature and science of sleep, 11, 301–315. Insomnia, Short Sleep, And Snoring In Mid-To-Late Pregnancy: Dispariti | NSS  

We agree that body mass index is implicated in many disorders related to sleep. Unlike the study above, where BMI is obtained from medical records, BMI as a variable is not available in the NRD dataset. Instead, elevated BMI is identified by ICD-9/10 codes. Although we cannot report or comment on directly in this study, the diagnosis code Z68.4 (and it’s ICD-9 equivalent) picks up obesity as part of the validated obstetric comorbidity index (Table 2).